# Nutraceuticals and Enteric Glial Cells

**DOI:** 10.3390/molecules26123762

**Published:** 2021-06-21

**Authors:** Laura López-Gómez, Agata Szymaszkiewicz, Marta Zielińska, Raquel Abalo

**Affiliations:** 1High Performance Research Group in Physiopathology and Pharmacology of the Digestive System (NeuGut), Department of Basic Health Sciences, University Rey Juan Carlos, 28933 Alcorcón, Spain; laura.lopez.gomez@urjc.es; 2Department of Biochemistry, Faculty of Medicine, Medical University of Lodz, 90-419 Lodz, Poland; agata.szymaszkiewicz@stud.umed.lodz.pl (A.S.); marta.zielinska@umed.lodz.pl (M.Z.); 3Associated Unit to Institute of Medicinal Chemistry (Unidad Asociada I+D+i del Instituto de Química Médica, IQM), Spanish National Research Council (Consejo Superior de Investigaciones Científicas, CSIC), 28006 Madrid, Spain; 4Working Group of Basic Sciences in Pain and Analgesia of the Spanish Pain Society (Grupo de Trabajo de Ciencias Básicas en Dolor y Analgesia de la Sociedad Española del Dolor), 28022 Madrid, Spain

**Keywords:** coffee, enteric glial cells, glia, inflammatory bowel disease, irritable bowel syndrome, neuropathic pain, nutraceuticals, quercetin, resveratrol

## Abstract

Until recently, glia were considered to be a structural support for neurons, however further investigations showed that glial cells are equally as important as neurons. Among many different types of glia, enteric glial cells (EGCs) found in the gastrointestinal tract, have been significantly underestimated, but proved to play an essential role in neuroprotection, immune system modulation and many other functions. They are also said to be remarkably altered in different physiopathological conditions. A nutraceutical is defined as any food substance or part of a food that provides medical or health benefits, including prevention and treatment of the disease. Following the description of these interesting peripheral glial cells and highlighting their role in physiological and pathological changes, this article reviews all the studies on the effects of nutraceuticals as modulators of their functions. Currently there are only a few studies available concerning the effects of nutraceuticals on EGCs. Most of them evaluated molecules with antioxidant properties in systemic conditions, whereas only a few studies have been performed using models of gastrointestinal disorders. Despite the scarcity of studies on the topic, all agree that nutraceuticals have the potential to be an interesting alternative in the prevention and/or treatment of enteric gliopathies (of systemic or local etiology) and their associated gastrointestinal conditions.

## 1. Introduction

The enteric nervous system (ENS) is a complex network of neurons and accompanying glial cells (enteric glial cells, EGCs) which controls the major functions of the gastrointestinal (GI) tract. 

At first, glia were considered to be just a structural support for neurons, but recent findings emphasized more on their functions, and they turned out to be equally as important as neural cells, due to their involvement in all aspects of neural functions for both the central and peripheral nervous system, including the ENS. 

Among the different types of glial cells (for example, astrocytes, microglia, Schwann cells), EGCs have been mostly underestimated, particularly regarding the modulation of their functions by nutraceuticals. However, EGCs are more often being recognized for their essential roles in physiology and the disease [1]. 

In this review, we focus on the enteric glia, their role and functions in physiology and pathology as well as the available studies on the effects of different nutraceuticals as modulators of these interesting cells. 

## 2. Enteric Glial Cells

The ENS is a network of neurons divided into submucosal and myenteric plexuses, together with their accompanying glia, the EGCs [2]. Originally, EGCs were considered as a structural support for the enteric neurons, however recently, it was proved that they are crucial for the functioning of the GI tract under physiological (intestinal barrier support, GI motility, sensation) and pathophysiological conditions (GI motility disturbances, visceral pain). 

Hanani et al. distinguished and classified EGCs into four subgroups based on their morphology (Table 1) [3].

Besides morphology, EGCs may also be classified according to the molecular or functional differences in receptors or channels expressed on their surface or in their nuclei. The following proteins are currently used to identify EGCs, i.e., calcium-binding protein S100β [4], glial fibrillary acidic protein (GFAP) [5] and the transcription factors: SOX8, SOX9, SOX10 [6].

As shown in Table 2, EGCs share some similarities with astrocytes, the major type of glial cells of the central nervous system. 

### 2.1. Cellular and Tissular Roles of EGCs

Generally, EGCs are considered as non-excitable cells, as they are unable to generate an action potential. Furthermore, EGCs are interconnected and electrically coupled by gap junctions that form an extensive glial network [9], as shown in Figure 1.

Enteric glial cells communicate with surrounding cells (neurons, glia, epithelial cells, immune cells) and integrate received information through calcium signaling [10]. Intercellular communication is a result of the propagation of calcium waves through connexin 43 (Cx43) hemichannels [11]. Moreover, EGCs are susceptible to the activation by neural pathways: intrinsic (from enteric neurons) or extrinsic (from autonomic or primary afferent neurons). The major neurotransmitter involved in this extracellular signaling is adenosine triphosphate (ATP) [12]. It was found that intermuscular EGCs express the purinergic receptor (P2X7) [13]. 

Like neurons, EGCs may release neurotransmitters and express the receptors for neurotransmitters on their surface to receive signals [13,14,15,16]. In particular, human EGCs were found to be immunoreactive to glutamate [17] and gamma amino butyric acid (GABA) transporter (GAT2) [18,19]. Furthermore, EGCs exhibit immuno-reactivity for L-arginine, a nitric oxide (NO) precursor and thus they may be involved in nitrergic neurotransmission [20,21].

Interestingly, EGCs are characterized by displaying a remarkable function: they may be activated upon stimulation (e.g., inflammation or following the injury), and switched into a reactive, pro-inflammatory phenotype [22,23]. When EGCs are activated, they have an increased ability to proliferate [24], enhance c-fos expression, and change their expression of markers and surface receptors [25]. For example, the expression of nerve growth factor (NGF) receptor, tropomyosin receptor kinase A (TrkA) [26], endothelin-1 receptor B (ET-B) [27], Toll-like receptor (TLR) 4 [28], and bradykinin receptor 1 (BR1) [29] are increased in enteric glia incubated with interleukin-1β (IL-1β), and TrkA receptor is up-regulated in response to lipopolysaccharide (LPS) stimulation [26]. 

Consequently, reactive glial cells are characterized by an increased expression of enteric glial markers. For instance, the expression of GFAP may be induced by the incubation with tumor necrosis factor α (TNF-α), IL-1β, LPS or LPS + interferon γ (IFN-γ) [27,30,31]. The latter also increases the expression of S100β [32]. In vivo, increased GFAP expression in the rat myenteric plexus occurred in LPS-induced intestinal inflammation [33]. 

Moreover, reactive EGCs are able to release neurotrophins, growth factors or cytokines and therefore enteric glia recruit immune cells (macrophages, neutrophils, mast cells) into the colonic mucosa [34,35,36]. This confirms an important immunomodulatory role for these cells within the GI tract.

The EGCs that are located directly underneath the epithelial layer constitute a link between the epithelium and submucosal neurons, and they participate in all steps of epithelial regeneration (cellular differentiation, migration, adhesion and proliferation) [37]. Therefore, EGCs support the epithelial barrier integrity in the intestines and have the capacity to enhance epithelial healing. Glial cell derived neurotrophic factor (GDNF) released by EGCs entails anti-inflammatory effect in the intestines through the inhibition of cellular apoptosis and decrease of pro-inflammatory cytokine level [35,38]. Furthermore, during mild inflammation, GDNF helps in the processes of epithelial reconstitution and maturation [39]. In addition, EGCs produce and release several factors involved in the processes of epithelial regeneration: pro-epithelial growth factor (pro-EGF) [40]. S-nitrosoglutathione [41] or 15-deoxy-Δ12,14-prostaglandin J2 (15d-PGJ2) [42]. EGCs support the intestinal barrier through decreasing intestinal permeability [41] or increasing the resistance to infections [43].

Enteric glial cells are also involved in the control of GI motility, as they coordinate sensory and motor signaling within the GI tract [44]. Noteworthily, according to Aubé et al. [45], a progressive loss of EGCs in transgenic mice, expressing haemagglutinin (HA), that received activated HA specific CD8+ T cells, led to the prolongation of the GI transit. In the study by Nesser et al. [46], in mice treated with fluorocitrate, a selective gliotoxin, the upper GI transit time was prolonged and the intestinal motility patterns were impaired (both the basal tone and the amplitude of contractility in response to electrical field stimulation were decreased).

Finally, ECGs are considered to be involved in visceral sensation, via directly or indirectly sensitizing or activating nociceptors. Additionally, EGCs have the potential to regulate nociceptor sensitization/activation by removing of neuromodulators [47]. Direct mechanisms of sensitization include the release of neuromodulators such as ATP, GABA, IL-1β and neurotrophins. Indirect mechanisms involve antigen presentation through major histocompatibility complex (MHC) class I and II, leading to activation of T cells followed by cytokine release, and regulation of other immune cells, leading to release of histamine and further cytokines (TNF-α, IL-1β) [47]. Moreover, pro-inflammatory signals induce glial Cx43-dependent macrophage colony-stimulating factor (M-CSF) production through protein kinase C (PKC) and TNF-α converting enzyme (TACE). This further supports the importance of EGC interaction with macrophages in the regulation of visceral hypersensitivity during chronic inflammation [48].

### 2.2. Physiological Changes in the Population of EGCs

The population of EGCs may be altered by many physiological factors, such as aging or diet modifications. The process of aging of the GI tract includes a progressive loss of EGCs. Philips et al. [49] compared the population of EGCs in the GI tract in young (5–6 months-old) and old (26-month-old) rats. According to their findings, there was a significant decline in the number and density of EGCs in the myenteric plexus from the duodenum up to the distal colon with age. However, there was a small, non-significant decrease of EGC number in the rectum.

Interestingly, more detailed research revealed that diet also influences the population of EGCs. A high-fat diet caused a significant loss of EGC density in duodenal submucosal plexus in mice [50]. On the contrary, the same diet increased the number of EGCs in the myenteric plexus of the antrum, while it remained unchanged in the jejunum [51]. In contrast to the alterations in the enteric glia, high-fat diet led to a substantial loss of myenteric neurons, while the population of submucosal neurons stayed within the norm [52]. 

On the other hand, food restrictions that slow the aging process by reducing the oxidative processes thus inhibiting of cell death, turned out to be detrimental for the EGCs. According to the study by Schoffen et al. [53], diet restriction accentuated morphologic and quantitative changes in glial cell populations in rats, whereas the 50% reduction of food supply entailed neuroprotective effects on the myenteric neurons in the colon. 

Nevertheless, the mechanisms responsible for the gliopathy occurring with age or diet modifications remain unknown. It is yet to be clarified whether the changes in morphology or number of EGCs are due to a direct impact of aging/diet restrictions or rather a consequence of the concomitant degenerative processes of the neurons in the ENS.

### 2.3. Role of EGCs in GI Pathophysiology

As EGCs coordinate the communication between the cells in the GI tract (neurons, epithelial cells, myocytes), any alterations in their population (such as those associated with the occurrence of different diseases) may have a significant impact on the GI functions.

#### 2.3.1. Intestinal Inflammation

Inflammatory bowel disease (IBD) is a group of chronic inflammatory conditions of the GI tract and two major types, Crohn’s disease (CD) and ulcerative colitis (UC) are distinguished. The first reports regarding the importance of enteric glia in the inflammatory processes in the GI tract came from 1998. Bush et al. [54] generated transgenic mice through the ablation of GFAP-positive glial cells from the jejunum and ileum, resulting in fulminating and fatal jejuno-ileitis. The ablation of EGCs led to severe inflammations, causing degeneration of neurons in the ENS and hemorrhagic necrosis of the small intestine. The alterations within the gut were similar to the pathology in the course of IBD in both animals and humans [54]. Consequently, the concept about the involvement of EGCs in the inflammatory processes in the GI tact emerges.

Noteworthily, Pochard et al. [25] summed up the results of molecular studies on the population of EGCs in IBD: in most studies, the expression of GFAP, S100β and GDNF was elevated in inflamed colon of IBD patients (both CD and UC) in comparison to their healthy colonic tissue [55,56,57]. The expression of GFAP was decreased in healthy intestinal samples from CD patients [55,56], but not UC, comparing to healthy patients. The expression of S100β was downregulated in the myenteric plexus of uninflamed areas from CD patients in comparison to healthy controls [57]. Likewise, in the rectum of UC patients the submucosal expression of S100β was decreased in comparison to healthy controls [23,30]. Noteworthily, GDNF production was increased in samples collected from healthy parts of the colon of UC patients as compared to healthy controls [56]. Interestingly, GDNF ameliorated experimental colitis, inhibited mucosal inflammatory response and decreased intestinal permeability in the mouse model of colitis induced by dextran sodium sulfate (DSS) [58]. 

The differences in the expression of glial markers in the course of IBD do not reflect the extent of alterations in the population of EGCs in the intestines during inflammation. The decreased expression of GFAP, located in the cytoplasm of CD patients may be considered as a sign of glial loss, but GFAP immunohistochemical staining is not optimal to quantify the number of cells. The emerging approach, that could possibly be used for further assessment of the enteric glia population in the course of IBD is the utilization of proteins located in the nucleus (such as SOX 8/9/10) [59]. 

Besides the potential glial loss in the course of IBD, the functional differences appear to be significant. Coquenlorge et al. [60] assessed that, although EGCs isolated from controls and CD patients exhibited similar expression of glial markers (GFAP, S100β) and EGC-derived factors (IL-6, TGF-β, pro-EGF and glutathione (GSH)), they differed in their influence on the intestinal barrier. Enteric glial cells from CD patients failed in supporting the intestinal barrier and the healing process opposite to those from healthy controls. This study was further expanded on the UC patients. It assessed how EGCs isolated from UC patients affect epithelial barrier of the intestines. It was confirmed that, unlike CD patient derived EGCs, EGCs from UC patients preserve intestinal permeability. The efficiency of the intestinal barrier was similar in co-culture with EGCs derived from UC patients and healthy controls [25].

Under physiological conditions, MHC class I receptors are expressed on the enteric glia, while MHC class II remain almost undetectable [61,62]. However, after the exposure to enteroinvasive *Escherichia coli*, the expression of MHC class II on the enteroglial cells is increased [63]. Moreover, the expression of MHC class II was significantly increased in CD patients in comparison to healthy controls, in which the expression of these receptors was very low or even absent [61,62].

#### 2.3.2. Chronic Constipation

Chronic constipation is a condition characterized by a lack of frequent bowel movements or difficulties of stool passage. Chronic constipation may be related to the organic barriers in the colon or rectum (i.e., tumor), neuronal/muscular impairment (i.e., dysmotility in Parkinson’s disease), post-infection (megacolon caused by Chagas disease) or idiopathic (idiopathic constipation). The results of clinical studies on the importance of EGCs in the control of GI motility indicate that a loss of enteric glia in the ENS may be associated with dysmotility (i.e., idiopathic constipation or infectious-related dysmotility). 

According to Bassotti et al. [64], who examined patients with constipation and collected samples from the ileum and colon, there was a loss of EGCs in these tissues. Notably, the decrease in the number of EGCs was accompanied by the reduction of enteric neurons density. Similar results were obtained in a group of patients with severe, intractable constipation that underwent colectomy with ileorectostomy, as they displayed a significant decrease in neurons, EGCs and interstitial cells of Cajal. Constipated patients had significantly more apoptotic enteric neurons in comparison to controls [65].

Noteworthily, the population of EGCs in the submucosal and myenteric plexus was significantly decreased in the colon of patients with severe constipation due to obstructed defecation refractory to medical treatment or biofeedback training. At the same time, the enteric neurons were reduced only in the submucosal plexus [66]. 

In the case of megacolon occurring in the course of Chagas disease (an infectious disease caused by *Trypanosoma cruzi*) and idiopathic megacolon, there was a remarkable reduction in the number of neurons and EGCs in the ENS in the colonic specimens collected during surgery. However, the differences in the population of EGCs were more pronounced in the group of patients with infectious megacolon [67].

#### 2.3.3. Postoperative Ileus

Postoperative ileus (POI) is a condition that may occur after surgery of the abdominal cavity or the outer abdomen, which is associated with GI motility impairment and results in inhibition of peristalsis and distension. Although the pathophysiology of POI remains unknown, recent studies indicate that EGCs maintain an important role in this process. Stoffels et al. [68] investigated the molecular mechanism of POI in mice. They determined that the blockage of the receptor for interleukin 1 (IL-1R) attenuated the POI. These receptors were found to be expressed on the surface of EGCs in the myenteric plexus. The activation of IL-1R in cultured EGCs promoted an inflammatory response through an increase in IL-6 and monocyte chemotactic protein 1 (MCP-1) levels, which may be an important step in the development of POI.

#### 2.3.4. Irritable Bowel Syndrome

Irritable bowel syndrome (IBS) is a chronic disease of the GI tract that manifests with recurrent abdominal pain accompanied by GI motility disturbances. This functional GI disorder may be classified as diarrhea-predominant IBS (IBS-D), constipation-predominant (IBS-C) or mixed IBS, when both diarrhea and constipation occur in an alternate manner (IBS-M). 

According to Lilli et al. [69] the immunoreactivity of S100β was significantly reduced in the colonic biopsies of IBS patients, independently of the IBS subtype (IBS-C, IBS-D, and IBS-M). Furthermore, the incubation of the rodent EGCs with supernatants from the mucosal biopsies from IBS-C patients reduced the cellular proliferation. Noteworthily, exposure of rat enteric glia with IBS-D and IBS-M supernatants impaired ATP-induced Ca^2+^ response of these cells. 

In some cases, one more type of IBS can be distinguished: the one followed by the bacterial, viral or parasitic infection of the GI tract (post-infectious IBS, PI-IBS) [70,71]. Notably, there were many attempts to elucidate the molecular mechanism that underlies PI-IBS, for example: hyperplasia of enterochromaffin cells, increased intestinal permeability or enhanced cytokine production [72,73]. Importantly, one of the proposed mechanisms of PI-IBS followed by *Clostridium difficile* infection involves EGCs. Toxin B produced by *C. difficile* evokes cytotoxic and pro-apoptotic effects on EGCs in vitro. This harmful impact of toxin B on enteric glia results from the disorganization of cytoskeleton, early cell rounding with Rac1 glucosylation, cell cycle inhibition and increased susceptibility to apoptosis induced by the pro-inflammatory cytokines (TNF-α and IFN-γ). Importantly, despite these direct effects of toxin B, it is important that EGCs which survive the detrimental action of toxin B, do not recover and their function is not restored (they exhibit persistent Rac1 glucosylation, disturbances in the cell cycle and low apoptosis rate) [74]. The long-term effects of *C. difficile* infection on EGCs network may be pivotal for GI homeostasis, as enteric glia coordinate cell-to-cell communication in the intestines [75]. 

The severity of visceral hypersensitivity in IBS patients may be associated with brain derived neurotrophic factor (BDNF), a protein described as crucial in the process of neuropathic and inflammatory pain. The level of BDNF was significantly elevated in the colonic mucosal biopsies from IBS patients and corresponded with the abdominal pain severity [76,77]. The high-affinity receptor for BDNF, tropomyosin receptor kinase B (TrkB), is expressed on the surface of EGCs [77]. Interestingly, the expression of this receptor, along with GFAP and substance P (SP), was increased in the colonic mucosa of IBS patients. It suggests that BDNF may play a key role in the occurrence of visceral hypersensitivity, i.e., in the course of IBS, through the interactions with EGCs. It was determined that the administration of fecal supernatants from IBS-D patients failed to induce visceral hypersensitivity in BDNF ± mice in contrast to wild type animals. In wild type animals, the pain threshold to colorectal distension after IBS-D fecal supernatant administration was significantly elevated when pretreated with a TrkB antagonist (TrkB/Fc). Noteworthily, the induction of visceral hypersensitivity evoked the up-regulation of the same proteins (TrkB, GFAP, SP) as in IBS patients in wild type animals, but not in the BDNF ± mice [78]. Overall, the fecal supernatant from IBS patients induced hypersensitivity that may involve a BDNF-TrkB signaling pathway. Thus, BDNF appear to act as a link between visceral hypersensitivity and EGC activation. 

The first step of the non-pharmacological management of IBS is diet modification. A dietary approach may involve the consumption of a low FODMAP products (diet low in fermentable carbohydrates). It was assessed that IBS patients have a higher *Firmicutes/Bacterioidetes* ratio and bacteria from the phylum *Firmicutes* are considered as a major source of the short chain fatty acid butyrate, a small molecule metabolite arising from symbiotic bacteria fermentation from insoluble dietary fibers [79,80]. The lower supply of fermentable carbohydrates alleviates IBS symptoms [81]. Furthermore, the butyrate enemas induce visceral hypersensitivity in animals tested. It was elucidated that butyrate-induced hypersensitivity is associated with the up-regulation of NGF on messenger ribonucleic acid (mRNA) and protein level thus EGCs are one of the major sources of NGF in the GI tract. Noteworthily, NGF was co-expressed with GFAP and the co-localized immunostaining area of NGF and GFAP was increased in the colon of rats that received butyrate-enema. Furthermore, it was reported that the secretion of NGF from EGCs in the colonic lamina propria was increased after the butyrate-enema [82].

#### 2.3.5. EGC and Pathophysiology Outer the GI Tract

Intestinal motility disfunction may also be a characteristic symptom of diseases outer the GI tract, for example neurodegenerative diseases, such as Parkinson’s disease (PD) or prion diseases. PD is a long-term, multi-system disease of the CNS, which is related to degeneration of the dopaminergic neurons. Besides the motor symptoms (rigidity, tremor, dyskinesia), the intrinsic aspect of PD is a dysfunction of the GI tract. Patients experience nausea, dysphagia, abdominal distension and constipation. Studies show that, in the colon of PD patients, there was an increased expression of glial markers (GFAP, S100β, SOX10), and this was accompanied by the elevation of pro-inflammatory cytokines (TNF-α, IFN-γ, IL-1β, IL-6) at the mRNA level. However, there was no correlation found between the expression of glial markers or the inflammatory indicators and the severity of disease or GI symptoms [83]. Likewise, according to Clairembault et al. [84], in the colonic biopsies from PD patients, there was a GFAP over-expression and a reduction in GFAP phosphorylation comparing to healthy controls. These results suggest that EGCs may be involved in the GI dysfunction observed in the course of PD, nevertheless further research is needed to understand the mechanism of this process. 

Prion diseases are progressive and fatal neurodegenerative conditions which are caused by spreading of pathological isoforms of cellular prion protein. This pathological process affects astrocytes in the CNS and EGCs in the GI tract [85,86]. However, it was assessed that the prion replication sites were found in the ENS prior to the replication in the CNS [87]. Thus, the enteric glia may be essential in prion neuroinvasion, as the GI system constitutes to the major exogenous prion protein entry site and acts as the starting point for the prions en route to the brain [87].

Finally, many systemic diseases may cause alterations in the GI function inducing many effects on the EGCs. For example: diabetes [88] or autoimmune diseases such as rheumatoid arthritis [89]. As shown below, some dietary components have proved beneficial in protection against EGC alterations in those diseases, thus favoring the restoration of GI altered functions.

### 2.4. Effects of Nutraceuticals on Enteric Glial Cells

Studies focused on the effects of nutraceuticals on EGCs are relatively scarce and include a variety of compounds with anti-inflammatory, antioxidant, or modulatory properties (Table 3).

Nutraceuticals with antioxidant properties have attracted a great amount of interest motivated by their ability to counteract the damaging action of free radicals produced during pathological conditions of the GI tract affecting neurons and glial cells. Among these compounds, the usefulness of the amino acid L-glutamine [90,91] and the tripeptide glutathione (GSH) [90] has been evaluated in streptozotocin-induced (STZ) type 1 diabetic rats. In diabetes, the production of free radicals is a part of the pathological process. L-glutamine is an amino acid that acts as a substrate in the formation of GSH, an important endogenous antioxidant. The oxidative stress that occurs during diabetes is associated with an increase in cellular reactive oxygen species (ROS), which act together with proteins, nucleic acids and lipids, damaging and inducing cell death by necrosis or apoptosis. In these studies [90,91], ileal whole-mount preparations were used for immunohistochemistry to evaluate myenteric neurons and EGC, immunoreactive (IR) to HuC/D and S100β, respectively, through cell counting and morphometric analysis. Inside the GI tract, both neurons and EGCs were affected by oxidative stress and the antioxidant activity of L-glutamine and GSH prevented damaging of EGCs [90,91]. In particular, 2% L-glutamine and 1% L-GSH were shown to exert gliotrophic effects that indirectly protected the myenteric neurons in the animal model of diabetes [90,91]. 

Likewise, the antioxidant properties of extracts obtained from some plants such as *Trichilia catigua* [92] or fungi such as *Agaricus blazei Murril* [93] were shown to alleviate the effects of oxidative stress on EGCs in preclinical models of diabetes and aging.

The ethyl acetate fraction from *Trichilia catigua* contains two flavonoids with antioxidant properties: procyanidin B2 and epicatechin. In STZ diabetic rats, myenteric EGCs IR to S100β showed an overall reduction in the number but increase in cell body area in jejunal whole-mount preparations [92]. According to the authors, this increase in the area of EGCs could be due to development of cellular edema, provoked by the same metabolic alterations that affect neurons, or due to a specific EGC mechanism triggered to compensate for the loss of neurons and to maintain homeostasis. Furthermore, diabetes produced changes in protein expression, measured by Western blot, with an increase in S100β and a decrease in GFAP production, accompanied by a decrease in cells co-expressing S100β and GFAP in the jejunum mucosa, as observed in cryostat sections. Consistently, EGCs contribute to regulation of the barrier function in the GI tract and loosing of its integrity allows the translocation of normally excluded contents to occur through the mucosa (such as microorganisms and antigens, initializing and perpetuating inflammatory disorders and tissue damage). Thus, the reduction in EGCs could possibly contribute to diarrhea or constipation in patients with diabetes. In addition, EGCs act as a communication channel between the ENS and the local immune system, and their alteration would affect the immunity of the diabetic patients. The administration of this extract not only prevented an increase in the area of the EGC, but also caused a reduction of S100β expression, probably due to the antioxidant effects of its’ flavonoids [92]. Thus, procyanidin B2 and epicatechin may be useful nutraceuticals helping to preserve GI health through their glioprotective effects. 

The aging process is related to the worsening of physical and metabolic processes associated with dysfunctions of the immune system and disorders of the energy metabolism, leading to oxidative stress and, therefore, to the formation of free radicals that, if not eliminated, may damage the cells. In this context, an aqueous extract obtained from the fungus *Agaricus blazei Murril* (cogumelo du sol) containing polyphenols, flavonoids and glutamate has been investigated using rats of 7-, 12- and 23-months-old [93]. This extract was beneficial to avoid any cell damage and mitigate the reduction in the number of neurons and glial cells in the ENS, as seen in whole-mount preparations from jejunum in which myenteric S100β-IR EGCs were analyzed. Although the exact mechanism of action remains unknown, antioxidant effects of this fungus could act directly by eliminating the free radicals generated by EGC. Alternatively, the extract could have an indirect action through the conversion of glutamic acid into glutamine, which is in turn a precursor of GSH, a molecule with well-known antioxidant effects, as already mentioned.

Brazil nut (*Berthonelletis excelsa HBK*) is a natural source of selenium (whose consumption improves the action of antioxidant enzymes in the colon), unsaturated fatty acids such as omega-3 and omega-6 (which are related to neuroplasticity associated with diet and reduce oxidative stress) and polyphenols such as ellagic acid (with neuroprotective action). In healthy rats, Brazil nut supplementation decreased gastric emptying and produced changes in GFAP immunofluorescent labeling, evaluated in microtome sections of the colonic wall [94]. However, these changes were dose-dependent but not linear, since 5% supplementation produced an increase in GFAP while 10% supplementation reduced it. The effect of Brazil nut on the EGCs may be due to different factors. Firstly, selenium is involved in calcium homeostasis and EGC communication occurs through calcium waves, thus influencing enteric neurotransmission and affecting the functions of the intestine. Minerals such as selenium could condition glial behavior and responses, affecting gastric emptying and GFAP expression [94]. Secondly, Brazil nut contains unsaturated omega-3 and omega-6 fatty acids. Omega-6 fatty acid derivatives such as 15d-PGJ2 are produced by EGCs and act as glial mediators, controlling the differentiation of intestinal epithelial cells trough peroxisome proliferator-activated receptor (PPARγ) [42] and exerting neuroprotective functions in the ENS. Interestingly, in EGC cultures with the use of molecular expression analysis (Western-blot, polymerase-chain reaction, PCR) and immunofluorescence labeling, 15d-PGJ2 was shown to be capable of activating nuclear factor erythroid-derived 2-like 2 (NrF2) [95] causing an increase in GSH synthesis in neurons and protecting against oxidative stress. Thus, exogenous sources of omega-6 derivatives such as 15d-PGJ2 (such as Brazil nuts) could constitute an interesting pharmacological or nutritional intervention in pathologies where EGCs are altered [95].

Coffee is one of the most popular beverages in the world and the beneficial effects of its components have been widely studied. Among them, caffeic acid (CA) and chlorogenic acid (CGA) are compounds with major antioxidant activity. In a mouse PD model, the administration of these compounds prevented the neurodegeneration caused by rotenone (an inhibitor of the mitochondrial complex I) in the dopaminergic neurons of the *substantia nigra* and in the enteric neurons [96]. In addition, in co-cultures of enteric neurons and glial cells, CA and CGA inhibited the neurotoxic effects of rotenone, which reduced the expression of metallothionein-1,2 (MT-1,2) and caused a loss of enteric neurons. MT is a low molecular weight protein rich in cysteine that binds to metals such as zinc, copper, and cadmium. This protein participates in detoxification as well as eliminating free radicals. The MT-1 and MT-2 isoforms are considered physiologically equivalent and intervene, among other functions, in the response to oxidative stress. Thus, these coffee components could protect the enteric neurons from the action of rotenone by triggering the antioxidant properties of EGCs [96].

Polyphenols are a type of plant-derived nutraceuticals with antioxidant properties, which contribute to different plant functions such as pigmentation and resistance to environmental stress or pathogens [109]. They have one or more hydroxyl groups attached to a benzene ring in their chemical structure and two types are distinguished: flavonoids such as quercetin, and non-flavonoids such as resveratrol.

Quercetin is a flavonoid naturally found in many foods such as onions, apples, broccoli, tea and red wine. Due to its antioxidant properties, the possible beneficial effects of quercetin have been studied in the field of diabetes by several authors [97,98,99]. As mentioned above, diabetes causes a reduction in the number of neurons and EGCs due to the oxidative stress associated with this disease, and supplementation with quercetin results in beneficial effects in the STZ-diabetic rat model. Using immunohistochemistry in whole-mount preparations, it was demonstrated that quercetin protected, at least partially, against the diabetes-induced loss and morphometric alterations of enteric neurons and glial cells from caecum [97], duodenum [98] and jejunum [99]. As mentioned above, EGCs participate in neuroprotection by expressing the S100β protein, which has neurotrophic activity, induces the growth of neuronal extensions, dendrites and axons and participates in neuronal survival. This protein is also involved in the regulation of specific intracellular signaling pathways such as calcium homeostasis, cytoskeletal stability, and induction of apoptosis. In addition, EGCs release numerous neurotrophic and antioxidant factors, including GSH and GDNF, which activate neuropeptide Y, another antioxidant system, thus protecting against neuronal death [99]. Indeed, the authors attributed the neuroprotective effect of quercetin to its ability to increase the expression of Nrf2 in EGCs, since the increased expression of Nrf2 increases GSH synthesis and promotes neuroprotection [98]. Summing up, the antioxidant properties of quercetin include protection of EGCs and allow them to exert their neuroprotective activity in the ENS in the context of diabetes.

The antioxidant and anti-inflammatory effects of quercetin have also been shown to be beneficial in alleviating the effects of oxidative stress and the production of cytokines provoked by rheumatoid arthritis [89]. This autoimmune and inflammatory disease destroys cartilage and bone tissue, but it is considered a multisystem disease that also affects other organs. The presence of autoantibodies increases oxidative stress and production of pro-inflammatory cytokines that cause cell damage, affecting the GI tract [89]. In rats, chronic rheumatoid arthritis, induced by the intradermal injection of complete Freund’s adjuvant of 5% heat-killed suspension of *Mycobacterium tuberculosis* into the right hind paw, reduced the number of neurons and EGCs in the submucosal and myenteric plexuses of the jejunum and caused mucosal inflammation. Due to the loss of EGCs, a decrease in the expression of GFAP and GNDF was observed, and there was also an increase in the expression of S100β that would activate immune and inflammatory processes. This increase in S100β corresponds to an increase of the neurons in the body area and EGCs in the myenteric plexus due to a compensatory effect to the reduction in the number of glial cells. Nevertheless, in the submucosal plexus, the EGCs reduced their area, which indicates a deleterious gliopathic effect induced by the disease, that results in a decrease of the enzymatic activity of these cells. Quercetin reversed all the effects, probably due to an elimination of free radicals that damage these cells, associated to its powerful antioxidant effects. In addition, quercetin also had anti-inflammatory effects on the mucosa, which were not found in ibuprofen [89]. However, the authors acknowledged that, in healthy animals, quercetin may result in pro-oxidant and toxic effects, depending on the dose and time of administration [110,111]. Thus, caution is needed to establish the right way of using of this nutraceutical, particularly for prophylactic purposes against inflammatory conditions.

Among the non-flavonoid polyphenols, resveratrol (3,5,4′-trihydroxy-stilbene) has been the object of great interest. It is a substance naturally present in high concentrations in the grape peel, red wine, peanuts, pistachios or chocolate and, unlike other antioxidants, high doses of resveratrol are well tolerated and not toxic [100]. This makes resveratrol a good candidate for the treatment of diseases related to oxidative stress such as diabetes or after episodes of ischemia-reperfusion, a process that usually occurs in several clinical situations (thrombosis, hernia, hemorrhagic and hypovolemic shock, infection, abdominal surgery) and in inflammatory diseases such as Crohn’s disease [101,102]. 

In a rat model of intestinal ischemia-reperfusion (I/R) injury caused by occluding the superior mesenteric artery for 45 min, followed by 7 days of reperfusion, resveratrol reversed the proliferation of EGCs promoted by I/R (seen as an increase in cells labeled with S100β and GFAP) and was neuroprotective for the myenteric neurons. The antioxidant action of this compound seems to be key to its protective action on EGCs, through the elimination of free radicals, and the regeneration of endogenous antioxidants, mediated by the increase of GSH levels and the activity of the antioxidant enzyme glutathione peroxidase [101]. Whereas free resveratrol prevented neuronal loss, glial proliferation, and reactive gliosis in the ENS, when loaded in poly(anhydride) nanoparticulate transport systems, its effects were not improved. Moreover, empty nanoparticles caused hepatotoxicity and promoted intestinal injury [102]. Although this first attempt was not successful, it will be certainly interesting to develop nanoparticles made of more suitable materials as a means to improve the oral bioavailability of resveratrol.

Resveratrol was also tested in the STZ-diabetic rat model [100], where an increase in GFAP labeling was detected using immunohistochemistry and fluorescence analysis on whole-mount preparations of the myenteric plexus from the three segments of the small intestine (ileum, jejunum and duodenum), as a consequence of a moderate reactive gliosis. Despite its neuroprotective effect after damage, the increase in GFAP is considered detrimental to regeneration because it constitutes an obstacle to prevent the establishment of contacts and normal neural circuits. Resveratrol treatment in diabetic animals did not alter EGC density, but significantly reduced GFAP immunoreactivity, indicating a reversion of such gliosis.

It has been postulated that some nutraceuticals could modulate EGC activity through the activation of the cannabinoid system. Cannabinoids and cannabinoid-related compounds, due to their antioxidant and anti-inflammatory activity, could be interesting to modulate the inflammatory processes in which these cells intervene. EGCs do not express the typical cannabinoid CB1 and CB2 receptors [112,113,114]. However, EGCs express the peroxisome proliferation activation receptor α (PPARα), a nuclear hormone receptor to which transcription-related ligands bind [113,114], and whose activation may have anti-inflammatory and antinociceptive effects [28]. The presence of this receptor in EGCs has led to the study of the action of ligands of this receptor such as palmitoylethanolamide (PEA).

PEA is a cannabinoid-related molecule involved in protective mechanisms. It is an endogenous fatty acid amide naturally found in foods such as egg yolks, peanuts, and soy lecithin, and is also produced endogenously. In a mouse model of DSS-induced colitis and in colonic biopsies from UC patients PEA produced anti-inflammatory effects. These were due to its ability to activate PPARα receptors and counteract glial activation, decreasing the production of NO and the expression of pro-inflammatory proteins such as inducible isoform of nitric oxide synthase (iNOS), cyclo-oxygenase-2 (COX-2) and TNF-α [28]. Similarly, in a rat model of human immunodeficiency virus (HIV)-induced diarrhea, produced by intracolonic administration of HIV-1 trans-activator of transcription (Tat) protein, PEA was shown to decrease this symptom of HIV-1 infection. The effects of PEA were mediated through the activation of PPARα receptors, present in EGCs, the consequent reduction of S100β and iNOS overexpression in the submucosal plexus of the colon and the blockade of TLR4/nuclear factor-kappa B (NF-kB) activation [103]. 

CBD is a non-psychotropic cannabinoid derived from the *Cannabis sativa* plant and a key mediator of glia-mediated neuroinflammation in the GI tract. The expression of the glial protein S100β has been studied in a mouse model of intestinal inflammation induced by LPS and in cultured rectal biopsies of patients with UC, after stimulation with LPS+IFN-γ [104]. S100β is related to inflammation, as it stimulates NO production. Thus, EGCs recognize inflammatory stimuli and once activated they produce and secrete S100β, which contributes to inducing iNOS, with the consequent production of NO. Pretreatment of mice with CBD prevented glial cell hyperactivation and decreased the expression of S100β and the infiltration of immune cells in the intestine, revealing its ability to prevent the amplification of the inflammatory response and S100β-mediated immune system activation. Similar results were obtained in the cultured rectal human biopsies, which were reversed by the PPARγ antagonist GW9662, confirming the involvement of this receptor in the anti-enterogliosis effect of CBD (although, to the best of our knowledge, this receptor has not been described so far to be expressed in EGCs). Thus, CBD may be useful in IBD patients, but high-quality clinical trials that may clearly determine the efficacy and safety of this compound, used as either a nutraceutical supplement or as part of medical cannabis, are still needed [115,116,117].

Another interesting nutraceutical for the treatment of UC is berberine, which is able to maintain intestinal EGCs and modulate the interaction between EGCs, epithelial cells and immune cells in EGCs and epithelial cell co-cultures [105]. Berberine is an isoquinoline alkaloid with anti-inflammatory effects present in different plants such as *Hydrastis canadensis, Berberis aquifolium* and *B. vulgaris*. During UC, changes occur in the expression of the neuropeptides secreted by the EGCs of the mucosa, such as GDNF or SP, which regulate functions such as inflammation, homeostasis or intestinal barrier function. Berberine administration improved intestinal damage in the DSS-mouse model of UC through restoration of the epithelial barrier, maintenance of resident EGCs and attenuation of immune infiltration and hyperactivation of immune cells. In vitro, berberine showed direct protective effects in EGC monocultures in a simulated inflammation situation and regulated the interaction between EGCs, epithelial cells and immune cells in co-cultures, further confirming its modulatory properties of mucosal inflammation [105].

Recently, the role of the microbiota has been demonstrated in various diseases, including diabetes [106]. The alteration of the intestinal microenvironment (dysbiosis) has been related to neurological disorders and alterations in EGCs. The recovery of the healthy microbiota through feeding using probiotic compounds can help in these cases, as some studies have shown [106,107,108]. Probiotics are defined as live microorganisms or bacteria that, administered in the diet, have beneficial effects on the health of the host [118]. Once ingested, they are potentially capable of influencing gastrointestinal functions due to their interactions with their components [107].

The levels of GDNF, GFAP and inflammatory markers such as IL-17, IL-6 and TLR-2, measured by enzyme linked immunosorbent assay (ELISA), were increased in the colon of type 2 diabetic rats. The elevation of these markers was related to both the dysbiosis, and the inflammation caused by diabetes that would activate glial cells as a means to prevent damage and protect neurons [106]. In this study, the application for 8 weeks after diabetes induction of *Lactobacillus plantarum* combined with the prebiotic fiber inulin (but not alone) improved the composition of the microbiota and reduced the levels of inflammatory cytokines and the expression of GDNF and GFAP [106].

In piglets, the administration of the probiotic *Pediococcus acidilactici* produced modifications in the ENS, as observed immunohistochemically in microtome serial sections [107]. The glial component was affected by the diet modification, but this modification was limited to the submucosal plexus of the ileum, with a significant increase in GFAP labeling, possibly related to ensuring the functional and structural integrity of the intestinal mucosa. In the ENS, neurons modify their chemical code in response to luminal changes and this neuronal plasticity aims to allow the adaptation of GI tract to changes or damage. Indeed, neurons may react to these alterations in the intestinal environment by modifying the expression of neuropeptides. EGCs participate in this neuronal plasticity through the secretion of neurotrophic factors. Thus, this study suggests that the changes in neurons and EGCs caused by the inclusion in the diet of the probiotic would be related to neuronal plasticity [107].

Finally, in a model of intestinal inflammation established by LPS and IFNγ- administration, the possible mechanisms by which two probiotics, *Bifidobacterium bifidum* (*B.b.*) and *Bacteroides fragilis* (*B.f.*), influence EGC regulation were examined [108]. In this study, it was observed that EGCs can increase the expression of inflammatory factors if they are subjected to harmful exogenous stimuli, and *B.b* can inhibit the inflammatory process by regulating EGCs, while *B.f.* does not have similar effects. Furthermore, EGCs are involved in intestinal inflammation and *B.b*. has protective effects by regulating EGCs. Both probiotics can influence the expression of NFG, neurotrophin-3 (NT-3), inflammasomes NLRP (nod-like receptor family, pyrin domain-containing) 3 and 6, IL-18, IL-1β and caspase-1 in EGCs, promoting or inhibiting intestinal inflammation. Both probiotics have beneficial effects by facilitating the elimination of microorganisms through the upregulation of inflammasome NLRP-3, leading to the inflammatory response, through the activation of caspase-1 and the secretion of IL-1 β and IL-8. Interestingly, *B.b.* increases the regulation of NLRP-6 mRNA while *B.f.* has inhibitory effects on its expression. NLRP-6 can inhibit the inflammatory response and plays an important role in maintaining homeostasis in the intestine. Interestingly, whereas *B.b.* decreased the expression of Il-18, IL-1β and caspase-1 in EGCs to thus inhibit the inflammatory response, *B.f.* showed opposite effects (i.e., it upregulated the expression of IL-18, IL-1β and caspase-1 in EGCs and promoted the inflammatory response). 

## 3. Conclusions

Here, we have reviewed the physiopathological roles of a relatively unstudied type of peripheral glial cells, i.e., enteric glial cells, as well as the research evaluating the effects of different nutraceuticals and food components on them. 

In general, studies on the modulatory effects that nutraceuticals exert on EGCs are relatively scarce, particularly those using models of GI diseases, such as IBD, in which PEA [28], CBD [104] and berberine [105] reduced inflammation, at least partly, through modulation of EGCs. PEA was also beneficial in a model of HIV-1 induced diarrhea [103], resveratrol protected the ENS against intestinal I/R [101,102], and the probiotics *Bifidobacterium bifidum* and *Bacteroides fragilis* inhibited and promoted, respectively, intestinal inflammation after LPS+IFN-γ administration [108]. 

The remaining studies were focused on systemic conditions that may also affect the GI tract. Curiously enough, most of them were performed in Brazilian institutions (Universidade Estadual de Maringá; Fluminense Federal University, Niterói), particularly those using the STZ-diabetic rat model and nutraceuticals with antioxidant properties, such as L-glutamine [90,91], L-glutathione [91], extracts of *Trichilia catigua* [92], quercetin [97,98,99] and resveratrol [100]. These Brazilian researchers have also evaluated the effects on EGCs of quercetin in a rat model of rheumatoid arthritis [92] and performed the mentioned study of resveratrol in a rat model of intestinal I/R [101,102]. Other Brazilian studies have been performed using extracts of the fungus *Agaricus blazei Murrill* in aged rats [93] and of Brazil nut in healthy animals [94]. Apart from these studies, a Japanese group (Okayama University) found beneficial effects of the antioxidant compounds caffeic acid and chlorogenic acid, from coffee, in the rotenone-induced mouse model of PD and also in cultures of EGCs exposed to rotenone [96]. Finally, some researchers from Italian and Iranian institutions have evaluated the effect of probiotics (*Pediococcus acidilactici, Lactobacillus plantarum*) and prebiotics (inulin) in healthy piglets [107] and rat models of type 2 diabetes [106]. 

Taken together, different nutraceuticals, particularly those with antioxidant activity, seem to exert neuroprotective effects in the ENS in local and systemic conditions that may involve not only direct actions on the enteric neurons but also indirect actions through EGC modulation. 

In the near future, we hope further studies will more precisely define the connections between nutraceuticals and EGCs as a possible target to treat, prevent or reduce their alterations associated with the different GI and systemic disorders in which they are involved. 

## Figures and Tables

**Figure 1 molecules-26-03762-f001:**
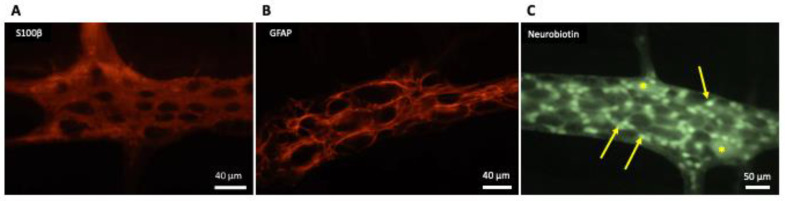
Appearance of enteric glial cells (EGCs). (**A**,**B**) Show the images obtained from the myenteric plexus of the rat distal colon; immunoreactivity to S100-β (**A**) and glial fibrillary acidic protein (GFAP) (**B**) are characteristic for EGCs. (**C**) Shows the network of electrically-coupled EGCs (arrows) in one myenteric ganglion from the guinea pig ileum; this image was taken as the result of the accidental insertion of an electrode filled with neurobiotin in one EGC while performing electrophysiological recordings of the activity of myenteric neurons (*); neurobiotin injected into one EGC diffused throughout the gap junctions connecting it with the other EGC in the myenteric ganglion, in the same way as firstly described by Hanani et al. [9] in 1989 for Lucifer yellow dye.

**Table 1 molecules-26-03762-t001:** Division of the enteric glial cells into subtypes.

Type	Morphology	Location
Type I Protoplasmic EGCs	Star-shaped cells with short, irregularly branched processes	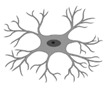	Intraganglionic (enteric ganglia)
Type II Fibrous EGCs	Elongated glia with branches	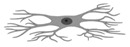	Within interganglionic fiber tracts
Type III Mucosal EGCs	Long branched processes	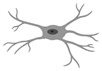	Extraganglionic: subepithelial glia
Type IVIntermuscular EGCs	Elongated glia	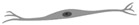	Extraganglionic: accompanying the nerve fibers and encircling the smooth muscles

Abbreviations: EGC, enteric glial cell. Created in biorender.com (accessed on 15 March 2021).

**Table 2 molecules-26-03762-t002:** Comparison between enteric glial cells and astrocytes [6,7,8].

Feature	Enteric Glial Cells	Astrocytes
Morphology	Irregularly branched processes	In vivo: numerous processes forming well-delineated bushy territories In culture: few processes, polygonal fibroblast-like shape Astrocytes show structural plasticity: their morphology differs between brain areas, and it may be changed (stellation or process growth) by different stimuli
Subtypes	Protoplasmic Fibrous Mucosal Intermuscular	Protoplasmic Fibrous
Location	Enteric nervous system (submucosal and myenteric plexus)	Central nervous system
Identification	GFAP Calcium-binding protein S100β Transcription factors (SOX8, SOX9, SOX10)	GFAP Calcium-binding protein S100β Glutamine synthase CD44 Vimentin Ran-2 Astrocytes from different brain regions can exhibit pronounced molecular differences
Adjacent cell coupling	Gap junction coupling	Gap junction coupling
Activation	Release of pro-inflammatory cytokines (i.e., IL-1β, TNF-α) Increased expression of c-fos, TrkA, ET-B, TLR-4, BR1 Enhanced expression of glial cell markers	Release of pro-inflammatory cytokines (i.e., IL-1β, IL-6, TNF-α, TGF-β) Increased expression of adhesion-related molecules (CD44) Increased expression of receptors for EGF, TNF-α Enhanced expression of glial cell markers: GFAP, vimentin, nestin
Involved in	Physiopathological modulation of GI functions	Development and plasticity of dendritic spines and synapses Elimination of dendritic spines, synapse formation Regulation of neurotransmission and plasticity

Abbreviations: BR1, bradykinin receptor 1; CD44, membrane glycoprotein; EGF, epidermal growth factor; ET-B, entothelin-1 receptor B; GFAP, glial fibrillary acidic protein; GI, gastrointestinal; IL, interleukin; Ran-2, rat neural antigen-2; TLR-4, Toll-like receptor 4; TGF-β, transforming growth factor β; TNF-α, tumor necrosis factor α; TrkA, nerve growth factor receptor.

**Table 3 molecules-26-03762-t003:** Effects of nutraceuticals on enteric glial cells.

Nutraceutical (Pathology)	Characteristics	Source	Effects	System	References
L-glutamine (Diabetes)	Amino acid	Protein-rich foods Available as a dietary supplement	Antioxidant	Wistar rats	[90,91]
L-glutathione (Diabetes)	Tripeptide	Supply of the raw nutritional materials used to generate GSH, such as cysteine and glycine	Antioxidant	Wistar rats	[90]
Phytotherapy (Aging, diabetes)	Procyanidin B2, epicatechin Glutamate, polyphenols Selenium, ellagic acid	*Trichilia catigua* *Agaricus blazei Murrill,* *Bertholletia excelsa H.B.,* (plants/fungi extracts and foods)	Antioxidant	Wistar rats	[92,93,94]
15d-PGJ2	Omega-6 fatty acid metabolite	Many sources (including *B. excelsa H.B.*)	PPARγ, Nrf2 activation	Non-transformed or transformed EGCs cultures	[42,95]
Coffee compounds (Parkinson’s disease)	Caffeic acid and chlorogenic acid	Coffee	Antioxidants Prevent MT-1,2 downregulation	C57BL/6 mice, EGC cultures	[96]
Quercetin (Diabetes, rheumatoid arthritis)	Flavonoid polyphenol	Fruits and vegetables Available as a dietary supplement.	Antioxidant, anti-inflammatory, Nrf 2 activation	Wistar rats, Holtzman rats	[89,97,98,99]
Resveratrol (Diabetes, intestinal ischemia-reperfusion)	Non-flavonoid polyphenol	Concentrated mostly in the skins and seeds of grapes and berries. Available as a dietary supplement.	Antioxidant, regulation of EGC proliferation	Wistar rats	[100,101,102]
PEA (IBD, HIV)	Endogenous or exogenous fatty acid amide	Soy lecithin, soybeans, egg yolk, peanuts, alfalfa. Available as a food-supplement named PeaPure.	PPARα activation, anti-inflammatory effects	Wistar rats, CD-1 mice, human intestinal biopsies.	[28,103]
Cannabidiol (UC)	Phytocannabinoid	*Cannabis sativa*	PPARγ receptor activation, anti-inflammatory effects	Mice, human biopsies	[104]
Berberine (UC)	Isoquinoline alkaloid	Plants from *Berberidaceae* family Available as a dietary supplement.	Modulation of interactions between EGCs, epithelial cells and immune cells.	C57BL/6 mice, Rat EGC cell line, CRL-2690 cultures	[105]
Probiotics (Diabetes, IBS)	*Lactobacillus plantarum* *Bifidobacterium bifidum* *Bacteroides fragilis* *Pediococcus acidilactici*	Fermented food and dairy products	Modulation of inflammation	Wistar rats, EGC cultures, pigs	[106,107,108]
Inulin (Diabetes)	Polysaccharide	Plants, also available in supplement form or as an ingredient	Prebiotic fiber	Wistar rats	[106]

Abbreviations: 15d-PGj2, 15-deoxy-Δ12,14-prostaglandin J2; EGC, enteric glial cell; HIV, human immunodeficiency virus; IBD, inflammatory bowel disease; IBS, irritable bowel syndrome; Nrf2, nuclear factor erythroid-derived 2-like 2; MT, Metallothionein; PEA, palmitoylethanolamide; PPARγ, peroxisome proliferator-activated receptor; UC, ulcerative colitis.

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
