# Peer review of "Nutraceuticals and Enteric Glial Cells"

_molecules, 2021, doi:10.3390/molecules26123762_

Round 1

Reviewer 1 Report

The work is well articulated and is exhaustive.

Author Response

In the review “Nutraceuticals and peripheral glia: focus on enteric and satellite glial cells” the

autors describes the physiopathological roles of two relatively unstudied types of peripheral glial cells, i.e., enteric glial cells (ECG) and satellite glial cells (SCG); they reported the beneficial effects on ECGs and SCGs of the nutraceuticals and food components.

The content of this work is well articulated and is exhaustive; for these reasons the work can be accepted in this form.

REPLY: We would like to thank the reviewer for his positive consideration towards our manuscript. We have improved it along the lines suggested by all the reviewers and hope the reviewer finds the new version of our manuscript satisfactory.

Reviewer 2 Report

Lopez-Gomez et al. surveyed and summarized literature on two relatively understudied PNS glial cell types: enteric glial cells (EGCs) in the gastrointestinal tract and satellite glial cells (SGCs) in the spinal cord. The authors focused on the responses of those glia to various pathological conditions as well as nutraceutical applications. Although a decent number of literature was included in the review, little insight was provided to critically evaluate the existing body of data, reach new conclusions, as well as identify gaps and potential research directions in the field. Overall, it is mainly descriptive by listing available observations without offering systematic viewpoints on the topic.

Major points:

  • The rationale of summarizing EGCs and SGCs is unclear except that both of them are understudies. In addition, there is no link between the two parts. The authors should consider to focus on one glial cell type and give up the other to make the whole review more coherent.

  • The authors should also consider to reframe the structure and content of the review. For instance, characteristics of EGCs and SGCs should be provided in the introduction or a separate section, where questions such as what kinds of proteins/receptors/channels are expressed by EGCs and SGCs, and how different are EGCs and SGCs from other astrocytes should be addressed. This information will provide necessary background to guide the readers through and place the following sections in context. Schematic drawings of EGCs and SGCs with their locations, morphologies and molecular profiles will be helpful.

  • There are numerous typos, grammatical errors and sentences that are long and confusing. While some sections are disproportionally long, important details regarding experimental methods and techniques are missing. The authors should substantially edit the article to make it more readable.

  • The objective of a review article is to provide a comprehensive understanding on a topic with knowledge/information as well as insights/evaluation. Unfortunately, the latter is missing.

Minor points:

  • In instruction, the sentence “Whereas microglia and Schwann cells have been intensely studied, satellite glial cells (SCGs) and enteric glial cells (ECGs) have received much less attention…” is odd. While all named glial cell types are from the PNS, microglia are from the CNS.

  • It would be more intuitive to present example images of different EGC types instead of describing the features in Table 1.

  • It seems contradictory that the authors presented findings where the numbers of EGCs were altered but then commented that “…, but currently there is no immunohistochemical staining that would enable to quantify the number of cells.” (In 2.3.1.). Please clarify.

Author Response

REVIEWER 2

Lopez-Gomez et al. surveyed and summarized literature on two relatively understudied PNS glial cell types: enteric glial cells (EGCs) in the gastrointestinal tract and satellite glial cells (SGCs) in the spinal cord. The authors focused on the responses of those glia to various pathological conditions as well as nutraceutical applications. Although a decent number of literature was included in the review, little insight was provided to critically evaluate the existing body of data, reach new conclusions, as well as identify gaps and potential research directions in the field. Overall, it is mainly descriptive by listing available observations without offering systematic viewpoints on the topic.

REPLY: We would like to thank the reviewer for his suggestions that have helped us to greatly improve our manuscript. We have addressed all issues posed by the reviewers and hope our manuscript is now satisfactory.

Major points:

  • The rationale of summarizing EGCs and SGCs is unclear except that both of them are understudies. In addition, there is no link between the two parts. The authors should consider to focus on one glial cell type and give up the other to make the whole review more coherent.

REPLY: Thank you very much for your suggestion. We decided to focus on the enteric glial cells, which are closer to the field of our interests (gastrointestinal system). 

  • The authors should also consider to reframe the structure and content of the review. For instance, characteristics of EGCs and SGCs should be provided in the introduction or a separate section, where questions such as what kinds of proteins/receptors/channels are expressed by EGCs and SGCs, and how different are EGCs and SGCs from other astrocytes should be addressed. This information will provide necessary background to guide the readers through and place the following sections in context. Schematic drawings of EGCs and SGCs with their locations, morphologies and molecular profiles will be helpful.

REPLY: Thank you for this comment and the suggestions for improvement. As mentioned above, we decided to focus on the enteric glial cells. Therefore, no comparison between enteric glial cells and satellite glial cells is provided in the new version of our manuscript. However, we agree with the reviewer that it is important to clarify the differences between our chosen population of peripheral glial cells and the main population of glial cells in the central nervous system, i.e., astrocytes. Therefore, we have now included a table (Table 2, pages 2-3, lines 65-85) that summarizes the main similarities and differences between these two types of glial cells regarding morphology, subtypes, location, molecular identification, coupling, factors involved in activation and main physiopathological roles. Furthermore, Table 1 (page 2, lines 55-56) now summarizes the types of enteric glial cells according to their morphology (a schematic drawing accompanies now the description of each morphological type) and location. We hope these two tables now clarify the points highlighted by the reviewer.

  • There are numerous typos, grammatical errors and sentences that are long and confusing. While some sections are disproportionally long, important details regarding experimental methods and techniques are missing. The authors should substantially edit the article to make it more readable.

REPLY: Thank you very much for your comment. We have provided more details on experimental methods and techniques used in the evaluation of the nutraceuticals on enteric glial cells (see section 2.4. Effects of nutraceuticals on enteric glial cells, pages 9-14, lines 370-649). Furthermore, according to the reviewer’s suggestion, we have reviewed the text for English correction. We have also tried to improve the readability of our manuscript by avoiding the use of excessively long sentences. We hope the new version of our manuscript is now more attractive to the readership.

  • The objective of a review article is to provide a comprehensive understanding on a topic with knowledge/information as well as insights/evaluation. Unfortunately, the latter is missing.

REPLY: Thank you for your comment. We totally agree. Therefore, we have tried to address this issue by critically evaluating the observations collected from the literature available, particularly those regarding the main objective of our review, the effects of nutraceuticals on enteric glial cells and their potential impact on the gastrointestinal tract functions (section 2.4. Effects of nutraceuticals on enteric glial cells, pages 9-14, lines 370-649). We hope the reviewer finds the new version of our manuscript more insightful, as required.

Minor points:

  • In instruction, the sentence “Whereas microglia and Schwann cells have been intensely studied, satellite glial cells (SCGs) and enteric glial cells (ECGs) have received much less attention…” is odd. While all named glial cell types are from the PNS, microglia are from the CNS.

REPLY: Thank you for your comment. It has been corrected, as needed.

  • It would be more intuitive to present example images of different EGC types instead of describing the features in Table 1.

REPLY: Thank you for your comment. As mentioned in our reply to a previous comment, schematic drawings of the different EGC types have now been included in Table 1 (page 2, lines 55-56).

  • It seems contradictory that the authors presented findings where the numbers of EGCs were altered but then commented that “…, but currently there is no immunohistochemical staining that would enable to quantify the number of cells.” (In 2.3.1.). Please clarify.

REPLY: Thank you very much for highlighting this issue. The immunofluorescence measure of glial markers (GFAP or S100β) does not always work well for counting the number of EGCs but generally allows the observation of the network of these cells in the ganglia (like in Figure 1, page 3, lines 90-96). Furthermore, the expression of these molecules is modified depending on the level of activity of enteric glia (for example, expression may be increased under inflammatory conditions, without this necessarily being translated into an alteration in the number of EGCs). As a matter of fact, Hoff et al. (2008) debated that previous research truly answered the question whether the number of glial cells is affected in the course of disease and suggested that other markers, found in EGC nucleus (namely, Sox 8/9/10), may be more useful to this aim. To clarify the issue, we have now modified the sentence cited by the reviewer as (page 6, lines 221-226): “The decreased expression of GFAP, located in the cytoplasm in CD patients may be considered as a sign of glial loss, but GFAP immunohistochemical staining is not optimal to quantify the number of cells. The emerging approach, that could be applied for further assessment of the enteric glia population in the course of IBD may be the utilization of proteins located in the nucleus (such as Sox 8/9/10)”. We hope this is clarified now.

Reference:

  • Hoff S.; Zeller F.; von Weyhern CWH.; Wegner M.; Schemann M.; Michel K.; et al. Quantitative assessment of glial cells in the human and guinea pig enteric nervous system with an anti-Sox8/9/10 antibody. J Comp Neurol. 2008, 509, 356–71.

Reviewer 3 Report

This is an interesting review that summarizes the studies on the effort of nutraceuticals on peripheral glia, enteric glial cells (EGCs), and satellite glial cells. The review on the basic physiology and pathology of these glial subtypes are up-to-date that is important for the readers. The information summarized in tables 2 and 3 is very informative to the field. 
Overall, the review is well-written and easy to follow.

Minor issues.
Line 33. add another type of glia in the CNS – oligodendrocyte precursor cells (OPCs) or NG2 glia.
Line,85. Unclear what is the “immune-reactivity to glutamate”?
Line 618: should elaborate more with respect to the similarity of satellite glial with Schwann cells.
Line 629, interesting note on the extracellular spaces between neuron and SGCs. Any information about the contact of SGCs with blood vessels. This information would be valuable to compare this PNS glia with astrocytes. 

Author Response

REVIEWER 3

This is an interesting review that summarizes the studies on the effort of nutraceuticals on peripheral glia, enteric glial cells (EGCs), and satellite glial cells. The review on the basic physiology and pathology of these glial subtypes are up-to-date that is important for the readers. The information summarized in tables 2 and 3 is very informative to the field. 
Overall, the review is well-written and easy to follow.

REPLY: Thank you very much for your positive consideration towards our manuscript. We have addressed all the points risen by the reviewers in order to improve our manuscript. We hope it is now satisfactory.

Minor issues.

Line 33. add another type of glia in the CNS – oligodendrocyte precursor cells (OPCs) or NG2 glia.

REPLY: Thank you very much for your comment. The introduction has been re-arranged, so that this part is no longer included. We hope this is acceptable.

Line 85. Unclear what is the “immune-reactivity to glutamate”?

REPLY: Thank you for your comment. We meant that enteric glia, so as neurons in the PNS and CNS and glia in the CNS, express glutamate, which is involved in bi-directional neuron-glia communication. In the reference provided (Giaroni et al, 2003), it was demonstrated that EGCs are immunoreactive to glutamate, stated as follows: “Fig. 2. Human colonic myenteric ganglia. Glutamate immunoreactivity is present in neurons (arrow) and in glial cells (arrowhead). Calibration bar: 50 Am”; “The present study shows that glutamatergic neurons are present in the human ENS. In agreement with the previous studies carried out in the guinea-pig and rat small intestine (Liu et al., 1997), glutamate immunoreactivity was found in neurons both in myenteric and submucosal ganglia as well as in axons innervating the circular muscle layer of the human large intestine. At this level, glutamate immunoreactivity was also found in enteric glial cells”. The text now reads (page 4, lines 107-108): “In particular, human EGCs were found to be immunoreactive to glutamate”. We hope this is now satisfactory.

References:

  • Giaroni C.; Zanetti E.; Chiaravalli AM.; Albarello L.; Dominioni L.; Capella C, et al. Evidence for a glutamatergic modulation of the cholinergic function in the human enteric nervous system via NMDA receptors. Eur J Pharmacol. 2003, 476, 63–9.
  • Liu MT.; Rothstein JD.; Gershon MD.; Kirchgessne AL.; Glutamatergic Enteric Neurons. J Neurosci. 1997, 17, 4764-4784.

Line 618: should elaborate more with respect to the similarity of satellite glial with Schwann cells.

REPLY: Thank you for this suggestion. Since the review has now been focused on the enteric glial cells, this comparison between satellite glial and Schwann cells was not made. This will be approached in future studies. We hope this is acceptable.

Line 629, interesting note on the extracellular spaces between neuron and SGCs. Any information about the contact of SGCs with blood vessels. This information would be valuable to compare this PNS glia with astrocytes. 

REPLY: Thank you for this suggestion. Since the review has now been focused on the enteric glial cells, no information regarding extracellular spaces between neurons and satellite glial cells or regarding the contact of SGCs with blood vessels has been included. This will be approached in future studies. We hope this is acceptable.

Round 2

Reviewer 2 Report

The authors have addressed all my concerns and questions.